# Geodesic Path Model for Indoor Propagation Loss Prediction of Narrowband Channels

**DOI:** 10.3390/s22134903

**Published:** 2022-06-29

**Authors:** Abdil Kaya , Brecht De Beelde, Wout Joseph, Maarten Weyn, Rafael Berkvens

**Affiliations:** 1IDLab-IMEC, Faculty of Applied Engineering, University of Antwerp, Sint-Pietersvliet 7, 2000 Antwerp, Belgium; maarten.weyn@uantwerpen.be (M.W.); rafael.berkvens@uantwerpen.be (R.B.); 2WAVES-IMEC, Department of Information Technology, Ghent University, Technologiepark-Zwijnaarde 126, 9052 Ghent, Belgium; brecht.debeelde@ugent.be (B.D.B.); wout.joseph@ugent.be (W.J.)

**Keywords:** radio channel, path loss, signal strength, receivers, transmitters, wireless communication, computational modeling, path planning, electromagnetic propagation, loss measurement, propagation loss

## Abstract

Indoor path loss models characterize the attenuation of signals between a transmitting and receiving antenna for a certain frequency and type of environment. Their use ranges from network coverage planning to joint communication and sensing applications such as localization and crowd counting. The need for this proposed geodesic path model comes forth from attempts at path loss-based localization on ships, for which the traditional models do not yield satisfactory path loss predictions. In this work, we present a novel pathfinding-based path loss model, requiring only a simple binary floor map and transmitter locations as input. The approximated propagation path is determined using geodesics, which are constrained shortest distances within path-connected spaces. However, finding geodesic paths from one distinct path-connected space to another is done through a systematic process of choosing space connector points and concatenating parts of the geodesic path. We developed an accompanying tool and present its algorithm which automatically extracts model parameters such as the number of wall crossings on the direct path as well as on the geodesic path, path distance, and direction changes on the corners along the propagation path. Moreover, we validate our model against path loss measurements conducted in two distinct indoor environments using DASH-7 sensor networks operating at 868 MHz. The results are then compared to traditional floor-map-based models. Mean absolute errors as low as 4.79 dB and a standard deviation of the model error of 3.63 dB is achieved in a ship environment, almost half the values of the next best traditional model. Improvements in an office environment are more modest with a mean absolute error of 6.16 dB and a standard deviation of 4.55 dB.

## 1. Introduction

Radio channel models characterize radio propagation for a certain frequency and type of environment and are valuable for the design of wireless communication systems. Path loss (PL) models characterize signal attenuation between a transmitting (TX) and receiving (RX) antenna and allow signal strength prediction and coverage calculations during the network planning phase [1], but can also be used for sensing [2] and localization [3] applications.

### 1.1. Related Work

In the past decade, a lot of research has focused on creating path loss models for indoor environments.

A recent and comprehensive survey on specifically indoor propagation models channel models is provided in [4]. The distance losses of the empirical or site-specific models discussed in this survey paper either use the direct path [5,6]. Geometric information about the environment is used in [6,7], but not without the use of explicit knowledge on the type of walls.

In [8], propagation measurements are presented in a corridor and office environment for frequencies ranging from 8 GHz to 11 GHz. In [9], models are created based on measurements up to 22 GHz in a corridor environment, for different angles of arrival and different antenna heights. Parameters are added to the close-in free space reference distance model and the floating-intercept model to better serve the ability to tune parameters. At such high frequencies, wall penetrations are not often taken into consideration. In [10], the linearity of attenuation due to obstructions or lossy media is discussed, based on measurements at 858 MHz and 1.935 GHz. The authors implicitly account for wall losses, under the assumption that the average wall loss in a particular environment is known, but instead of explicitly incorporating the number of walls crossing between transmitting (TX) and receiving (RX) antennas, an additional parameter is used, defined based on general information about the environment. An indoor path loss model for wireless local area networks accounting for wall attenuation is presented in [11]. The authors derive their model from the Average Wall Model. Instead of using the average wall attenuation within the environment, however, their model considers the different types of walls on the direct path between TX and RX and then averages the attenuation based on the wall crossed. This approach requires the wall types on a floor map and does not estimate non-direct propagation paths.

In [12], indoor path loss measurements in a residential living room at 60 GHz are presented. Radio propagation in office environments is well studied [8,13,14,15,16,17,18,19,20]. On the other hand, only limited literature is available for path loss models on board of naval vessels [21,22,23]. Previously, geodesic paths were used for ear-to-ear propagation in the 2.4 GHz frequency band by the authors of [24]. The distances considered in this work are at least two orders of magnitude smaller and are intended to consider geodesics without interruptions in path-connected space.

Aforementioned models and methods make no distinction in path-connectedness of the TX and RX antennas. Models and algorithms that are based on path finding are presented in [25,26,27]. The methodologies herein are based on finding the dominant path between a transmitter and a receiver. This dominant path is defined as the least attenuation accumulating path from transmitter to receiver. In order to do this, every possible path must be considered. The paths are considered consecutive connections between concave corners and center points of walls on a floor map. The connector points are used in a tree to search for a single path.

In this research, we present a novel path loss model for predicting signal strength based on a 2-dimensional binary floor map (representing walls and empty space, but not their types). The PL modeling approach is validated using experimental PL measurements in the cabin environment of a freight ship and in an office environment.

### 1.2. Background

In this section, we provide an overview of the different PL modeling approaches that are typically used and in terms of the required input, i.e., a floor map, best compared to our proposed model. We provide an implementation of the floor-map-based models under the constraint that floor maps do not make a distinction between wall types. It is often the case in real environments that a binary floor map is readily available, but wall types are not.

#### 1.2.1. One-Slope PL Model

The one-slope floating-intercept (FI) model from (Equation 1) describes a linear relation between the logarithmic distance and PL, with PL${}_{0}$ the PL in dB at reference distance d${}_{0}$ = 1 m, n the PL exponent and $d_{d}$ the Euclidean distance in meter between the TX and RX antennas.
(1)$${\mathrm{PL}}_{\mathrm{FI}}\left(d_{d}\right)={\mathrm{PL}}_{0}+n\;\;\cdot \;10\log_{10}\left(\frac{d_{d}}{d_{0}}\right)$$
The model parameters PL${}_{0}$ and n depend on the frequency and environment, and are fitted based on measurement data.

#### 1.2.2. Floor-map-based Path Loss Prediction

Several approaches exist to predict PL based on the floor map of the environment, taking into account attenuation due to objects obstructing the Line-of-Sight path, as well as floor and wall penetration.

The average wall model (AWM) from (Equation 2), also known as the Keenan-Motley model is a simplified model that only adds floor and wall attenuation to the free space PL model without distinguishing between different wall types, and without accounting for non-linearity of floor losses. In (Equation 2), $\bar{L_{w}}$ represents the average value for attenuation due to a wall crossing, and $k_{w}$ is the number of walls crossed on the direct line between TX and RX.
(2)$${\mathrm{PL}}_{\mathrm{AWM}}={\mathrm{PL}}_{0}+n\;10\log_{10}\frac{d_{d}}{d_{0}}+\bar{L_{w}}\;k_{w}$$

The COST 231 multi-wall model from (Equation 3) uses a free space PL model to which losses due to wall and floor intersections are added [28]. In this equation, $d_{d}$ is the Euclidean distance between the antennas, f is the frequency in Hz, c = $3\times 10^{8}$ m/s is the speed of light, $L_{c}$ is a regression parameter representing a constant loss, typically close to zero, $k_{wi}$ is the number of walls of type *i* that are penetrated, $L_{\mathrm{wi}}$ is the loss coefficient for wall type *i*, and I is the number of wall types, $k_{f}$ is the number of penetrated floors with floor attenuation $L_{f}$, and b is an empirical parameter to adapt to the non-linearity of floor losses.
(3)$${\mathrm{PL}}_{\mathrm{COS}T}=20\log_{10}\left(4\pi d_{d}\frac{f}{c}\right)+L_{c}+\sum_{i=1}^{I}k_{\mathrm{wi}}L_{\mathrm{wi}}+k_{f}^{\left[\frac{k_{f}+2}{k_{f}+1}-b\right]}L_{f}$$

The recommendation P.1238-11 from the International Telecommunication Union (ITU) presents both a site-general and ray-tracing-based site-specific model [29]. We will consider the site-general model from (Equation 4), because it best compares to the model we propose in this paper in terms of the input requirements. While this site-general model does not consider wall crossings, it implies the wall losses by making a distinction between coefficients for Line-of-Sight (LOS) and non-Line-of-Sight (NLOS) scenarios. This means that the model also requires a floor map. In this equation, $d_{d}$ is the Euclidean distance between the antennas, f is the operating frequency in GHz, α is the path loss exponent, β is an offset value, similar to $PL_{0}$ in the one-slope PL model, γ is a coefficient related to the increasing transmission loss with frequency f.
(4)$${\mathrm{PL}}_{\mathrm{ITU}}=10\;\alpha \log_{10}\left(d_{d}\right)+\beta +10\;\gamma \log_{10}\left(f\right)$$

#### 1.2.3. Ray Tracing Based Models

In ray tracing algorithms, rays are launched for different azimuth and elevation angles, and interactions with the environment are determined. The drawback is the high computational complexity, as well as the requirement of having an accurate description of the environment [30,31].

#### 1.2.4. Pathfinding Based Models

The indoor dominant path prediction (IDP) model provides ray tracing accuracy at a significantly lower complexity. IDP searches for only a single path, i.e., the dominant path, which contributes to the most received power at the receiver antenna RX. The IDP model justifies the use of a single path between TX and RX because more than 95% of the contributed power is contained in 2–3 rays [25,26]. To find the dominant path, without the computational intensity of ray tracing, IDP initially limits the number of ‘passages’ that a radio wave can supposedly go through when TX and RX are not in LOS. It does so by relying on a graph that connects distinct rooms by connector points in the center of shared walls. We refer to rooms as path-connected spaces in the remainder of this paper. Non-convex path-connected spaces are connected similarly by placing connector points in the concave corners. In doing so, the model considers an exhaustive list of all possible paths from TX to RX. A pre-defined heuristic then decides on the least loss-inducing path. Parameters along this path are then used to determine wall losses, diffraction losses, reflection loss, waveguiding, and distance losses. While the original IDP model used a neural network to find the path losses, the authors of [27] provide a thoroughly documented version of IDP with an intuitive model equation instead of a neural network. Even though IDP is conceptually similar to our proposed model, it requires detailed information such as the type of walls on the floor map to populate crucial reflection, diffraction, and waveguiding parameters. For this reason, we will not include IDP in the comparison and limit the comparison to floor-map-based models which can be used without wall type information.

### 1.3. Contributions

In this paper, we propose a path loss model and parameter estimation algorithm for path loss predictions. The proposed model only requires the input of a binary floor map, without the need for detailed wall type information. As opposed to the models in the previous sections, the proposed model is based on the combination of the direct path and the shortest geodesic path [32] between a transmitter and receiver. Instead of the euclidean distance, the distance along the points of the geodesic path is used. While the Euclidean distance of the direct link is not used as the distance metric, the number of wall crossings on the direct link is still used as a parameter. We find that in the proposed model, the impact of walls crossed on the direct link has a logarithmic relationship with the path loss, while the walls crossed on the geodesic paths (thus between distinct path-connected spaces) show a linear relationship with path loss. Using only the geodesic path, without performing ray-tracing, interaction losses are represented by the direction changes along the path. The model is implemented in a tool that automatically extracts the parameters of the model to estimate path loss based on a 2-dimensional floor map and predefined transmitter locations. The implementation is validated against path loss measurements using DASH-7 transceivers operating at 868 MHz in two different environments, i.e., the metallic environment of a ship and an office environment. This validation is then compared to three conventional floor-map-based path loss models. We evaluate the most commonly used floor-map-based models by tuning the coefficients to our measurement environments in our PL model tool. Given that the models compared range from site-general to site-specific and from zero to three tunable coefficients, the goal of the evaluation comparison is not to gauge a head-to-head performance difference, but rather to provide a set of broad, yet commonly used model performances in conjunction with our PL model tool. The compared results are however intended to contribute to an informed trade-off decision when selecting a PL model.

## 2. Methods

### 2.1. Path Loss Model

The proposed model in this paper is based on the approximated propagation paths of the radio signal. Many floor-map-based path loss prediction models either consider the direct path (AWM, COST 231, ITU-R P.1238) or choose a single most likely propagation path of the signal, discarding the direct path altogether (IDP). In the latter case, all the attenuation is implied to be a result of the approximated propagation path and the losses incurred along it. From our measurements, we find that even when considering an approximated propagation path, it is still useful to consider wall losses on the direct path with regard to the prediction of path losses. In our model, we make a distinction between walls crossed on the direct path and walls crossed on the propagation path. The former has a logarithmic relation to the path loss and the latter a linear one. We find and report on the importance of making a distinction between propagation paths crossing walls that divide path-disconnected spaces and those that cross walls that merely reside in the same path-connected space. We propose the geodesic path loss model (GPM) from (Equation 5). In this model, we use geodesics or shortest distances constrained by walls to determine the shortest paths within path-connected spaces. In the remainder of the paper, we specify the approximated propagation path as the geodesic paths. Finding geodesic paths can only be done in path-connected spaces, as such, we apply a systematic methodology to cross distinct path-connected spaces and concatenate multiple geodesic paths into one resulting propagation path. The extraction of GPM parameters is detailed in Algorithm 1. The coefficients and parameters of the model are as follows. PL${}_{0}$ is the reference path loss in dB at distance d${}_{0}$, n the path loss exponent, $d_{P}$ the path distance in meter of the (concatenated) geodesic path. $L_{\mathrm{wd}}$ and $L_{\mathrm{wp}}$ are the coefficients for the average wall losses on the direct path and geodesic path respectively. $k_{\mathrm{wd}}$ and $k_{\mathrm{wp}}$ are the number of walls crossed on the direct path and geodesic path respectively. $L_{\alpha }$ is the coefficient for the interaction loss expressed as the sum of direction changes $\alpha_{i}$ along the path.
(5)PLGPM=1PL01Lwd10log10kwd−kwp+Lwpkwp10nlog10dPd0︸︸Loss at 1 m+10nlog10dPd0Lwd10log10kwd−kwp+Lwpkwp︸ifLOS,Lα∑isin2αi210nlog10dPd0Lα∑isin2αi2︸Distance loss+Lwd10log10kwd−kwp+LwpkwpLα∑isin2αi2︸Wall losses+Lα∑isin2αi2︸Interaction lossesotherwise.10nlog10dPd0Lwd10log10kwd−kwp+Lwpkwp︸ifLOS,10nlog10dPd0Lα∑isin2αi2︸Distanceloss+Lwd10log10kwd−kwp+LwpkwpLα∑isin2αi2︸Wall losses+Lα∑isin2αi2︸Interaction lossesotherwise.

The subtraction $k_{\mathrm{wd}}-k_{\mathrm{wp}}$ in the logarithmic term serves to avoid the double counting of path-disconnected wall crossings $k_{\mathrm{wp}}$. If $k_{\mathrm{wd}}=k_{\mathrm{wp}}$, then the logarithmic term is removed.

A visual representation of the parameters described is shown in Figure 1.


**Algorithm 1:** Tool to determine geodesic paths and other PL model parameters
**Data:** Floor map $F\subset Z^{2}$, as a set of couples (2-tuple coordinates) $p_{i}\leftarrow \left(x_{i},y_{i}\right)$. *f* is a binary mapping of *F* into {0,1} such that f(p) is either 0 (open space) or 1 (walls). The set of wall coordinates W←{p∈F|f(p)=1}, antenna locations $p_{\mathrm{TX}}$ and $p_{\mathrm{RX}}\in \left(F-W\right)$.**Result:** Shortest (concatenated) geodesic path P, path distance $d_{P}$, direct distance $d_{d}$, the number of walls on the direct link $k_{wd}$, the number of walls on the geodesic path $k_{wp}$, an ordered list of direction changes across the geodesic path $\forall {}_{i}\alpha {}_{i}$ (rad).**Functions:** $\left[D_{p,C}\right]⟵GeoTransform\left(p,C\right)$: Returns the geodesic transform $D_{p,C}$ with seed point *p* in connected space (mask) *C*, using a quasi-euclidean 8-connected kernel, for which each element of $D_{p,C}$ represents a coordinate in *C* and its distance to *p*.$\left[P_{p,q,C},d_{P}\right]⟵GeoPath\left(p,q,C\right)$: Returns the geodesic shortest path from *p* to *q* in *C* by considering the thinned, minimal distance coordinates between *p* and *q*, resulting from the sum of the two geodesic transforms. $P_{p,q,C}⟵D\left\{D=\min\left(D_{p,C}+D_{q,C}\right)\right\}$. The shortest path distance is then equal to the value of any coordinate element in P. $d_{P}⟵P\left\{1\right\}$P⟵DouglasPeucker(P,ε): Returns the coordinate couples sequence of path P, by recursively decimating points which do not deviate more than tolerance ε from the current line segment under evaluation. This tolerance value is set to the max radius of the first Fresnel zone.


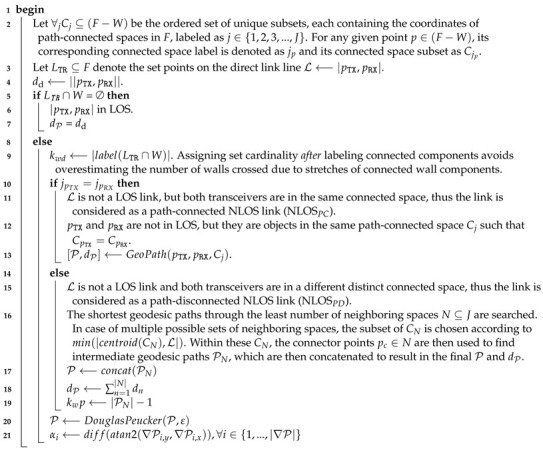




### 2.2. Model Implementation

Algorithm 1 presents the algorithm for our tool, used to determine PL parameters.

### 2.3. Experiment Sites and Setup

The proposed GPM is a site-specific model, just like the AWM and the COST 231 model. For these three models, the coefficients and terms need to be tuned or chosen on a per-environment basis. The ITU-R model is the only site-general model in the comparison and requires no tuning of coefficients. The coefficients are chosen based on the type of environment the path loss prediction is performed for.

The path loss measurements from two measurement campaigns are independently used to validate the different PL models. The first measurement environment, shown in Figure 2a, is the superstructure main cabin floor of a freight ship. The walls of this environment are all made of either steel for the superstructure’s load-bearing walls or aluminum for doors and thin compartmentalization walls. A second measurement environment, shown in Figure 2b, is a regular office floor in a 10-story building. The building materials of this environment range vastly from windowed walls to reinforced concrete. Typical materials such as timber and plasterboard for compartment walls, different types of metal alloys for the elevator shafts (near transceiver 17), and glass can be found in the environment. However, we explicitly do not take this information into account for any of the implemented models, as the assumption is that wall-type information is not available. Open doors are indicated on these two-floor maps by an actual opening, while closed doors are indicated as wall lines. Furthermore, we have not taken the thickness of the walls on the floor map into account. Any wall crossing is counted as such, a single wall crossing, with no distinction in wall types, even if the floor maps suggest differences in wall thickness.

To acquire the training and validation data from these environments, we held multiple measurement campaigns. The transceiver locations are shown in Figure 2. The specific link distinction between training and validation links are shown in Figure A2 of Appendix A and the number of links per environment are provided in Table 1. The deployed wireless sensor network (WSN) shown in Figure 2 is a highly connected DASH-7 WSN, operating at 868 MHz. The communication model, hardware, and network setup from the communication perspective are detailed in a previous work in [33]. The deployed transceivers on tripods, as well as the environment, are shown in Figure 3.

### 2.4. Model Parameter Algorithm Output

A small and random set of the resulting paths and parameters are shown in Figure 4. This shows the geodesic paths through path-connected spaces (NLOS${}_{\mathrm{PC}}$ links), path-disconnected spaces (NLOS${}_{\mathrm{PD}}$ links) as well as the corners and direction changes expressed in degrees (for illustration). Corners are determined using the Douglas-Peucker algorithm [34]. This algorithm requires a tolerance value, which we set at the maximum radius of the first Fresnel zone. In Appendix A, Figure A1, we provide a floor map with all measurement links drawn and represented as direct links. The algorithmically found geodesic paths are overlaid on this map.

### 2.5. Model Coefficients and Terms

Usually the path loss exponent n is either fitted to LOS measurements at a range of distances or set to the free space path loss (FSPL) exponent of 2. We prefer the latter, but because certain environments have a lower PL exponent than 2, such as highly metallic industrial environments or in this case a ship, both our own measurements as well as findings in literature indicate a PL exponent lower than 2 [1,35,36,37]. As such, in our case, we will use a PL exponent of n=2 for the office environment and n=1.15 for the ship environment. The reference path loss PL${}_{0}$ is set as 31.2 dB, which is the FSPL at an operating frequency of 868 MHz and 1 m distance from the TX antenna. The frequency dependent reference path loss term can be translated to other sub-6 GHz frequency bands. The frequency dependency thus stems from (Equation 6), but also from the corner selection process in which the maximum radius of the first Fresnel zone is used as the tolerance value in the iterative-end-point-fit algorithm.
(6)$${\mathrm{PL}}_{0}=20\log_{10}\left(d_{0}\right)+20\log_{10}\left(f\right)+20\log_{10}\left(\frac{4\pi }{c}\right)$$

All other model-specific parameters, unless prescribed by the model itself, were attained using multivariate nonlinear least squares. The GPM uses three coefficients to be fitted and both the AWM and the COST 231 models only one. The ITU-R model is not a site-specific model, but instead a site-general model, with predetermined coefficients and terms based on the type of environment, making it the only model that doesn’t need any coefficient fitting.

The number of unique measurement links between TX and RX is 117 for the ship environment and 273 for the office environment, with a total number of respectively 39,012 and 17,046 path loss measurements across those links. A third is used for coefficient tuning, presented in Table 2, while the remainder is used for model validation and error analysis. Table 2 does not include the coefficients for ITU-R P.1238, because those are pre-determined based on the type of environment and whether or not the current link between TX and RX is in LOS [29].

## 3. Results

After obtaining the various model coefficients, tuned against a training sample set of PL measurements, we evaluate the resulting predictions per model, per environment. First, we present the statistical error analysis. Paired with the PL prediction on every point of the maps, we can interpret and discuss the model behavior in conjunction with the error results with respect to the measured validation links.

### 3.1. Prediction Errors

Figure 5 shows different perspectives on how well the predictions from the different models fit. In Figure 5a,c, the PL is shown as a function of the direct distances.

For the model prediction comparisons for the ship environment in Figure 5a, we can see that the GPM predictions follow the measurements fairly well irrespective of the distance between TX and RX antenna, whereas the traditional floor-map-based models tend to have larger prediction errors for larger distances. The model prediction comparisons of the office environment in Figure 5c on the other hand show that predictions across all models tend to follow the measurements in a similar manner.

In Figure 5b,d, a scatter plot is shown. The *x*-axis corresponds to the measured PL on a communication link, with its respective PL prediction on the *y*-axis. This is done for two PL models, the GPM and AWM. The first bisector line represents a perfect prediction relation between the measured PL (dB) on the *x*-axis and the predicted PL (dB) on the *y*-axis.

In the office environment, AWM and COST231 are essentially the same model, given that, in this environment, the AWM uses FSPL PL${}_{0}$ and PL exponent n=2. The ITU-R P.1238 model (site-general version), which uses predetermined coefficients, makes predictions with very large errors of up to 41.72 dB as reported in Table 3.

Figure 6a,c show the empirical cumulative distribution functions (CDFs) for the absolute errors of the path loss predictions of all considered models. Similarly, Figure 6b,d show the CDFs, only now with the signed Errors, (PL${}_{\mathrm{PREDICTION}}-{\mathrm{PL}}_{\mathrm{MEASUREMENT}}$), such that the negative side of the *x*-axis represent the underestimations from the models and the positive side of the *x*-axis represents overestimations from the models.

Lastly, a summary of the statistical error analysis is provided in Table 3. A large disparity between GPM and other floor-map-based models is apparent in the results from the ship environment. The mean absolute error (MAE) is 4.79 dB for GPM, whereas that of the next best prediction model for the ship environment (ITU-R P.1238) is almost double at 8.79 dB. The standard deviation of the error is only 3.63 dB for the GPM. Standard deviations between 3 and 6 dB are excellent according to [38], and referred to in the works of [27]. Moreover, the coefficient of determination (R${}^{2}$) of the GPM is much better than that of the others. While the GPM outperforms other models in the office environment, the observed results are more comparable to each other. One exception is the ITU-R model, which under-performs in comparison to the site-general models in this specific office environment. While it is interesting to assess the performance of the ITU-R model, it remains a site-general model in which the coefficients are not tuned to the specific environment, unlike the other models. The GPM, AWM, and COST 231 models are trained, and their respective coefficients are tuned from the same training set. Since these three are site-specific models, the tuning is done separately for the ship and office environment, but equally within each environment.

### 3.2. PL Prediction Results

From the environment parameters and model coefficients, we predict the PL at every location on the floor map from a given TX antenna for both the ship and office environment. The presented prediction is limited to the GPM and AWM. The apparent differences between these two models are instructive and relay the shortcomings of traditional floor-map-based PL models, which we discuss in Section 4. Figure 7 provides an overview of the simulations.

## 4. Discussion

Originating from the need for a more representative PL model for the localization of transceivers in an indoor ship environment, we set out to find a pathfinding-based PL model. The goal of this model is to alleviate the systematic errors induced by map-based models that ignore propagation paths beyond the direct path, without requiring additional information such as wall types or predetermined connector points between rooms and around concave corners. The only input required for the model is a simple binary floor map, without any additional details about wall types. Such detailed information is seldom readily available, even when computer-aided design files are present. The model algorithm to find the parameters is part of the model. The algorithm is based on finding shortest paths using geodesics with a quasi-Euclidean kernel, the crossing between path-disconnected spaces using heuristics, and lastly, the finding of angles on corners, which are first determined by path point reduction using the Douglas–Peucker algorithm.

### 4.1. Experimental Validation

In Table 2, the GPM coefficients are presented along with the 95% confidence interval bounds for both the ship and office environment. Unsurprisingly, we see that for the ship environment, the impact of wall crossings between path-disconnected spaces has a very large impact, as indicated by $L_{\mathrm{wp}|GPM}$. Conversely, the impact of wall crossings across the direct path has a significantly lower impact, as indicated by $L_{\mathrm{wd}}$. However, the accumulated impact thereof is still sizable, because the number of walls crossed on the direct path is approximately ten times the number of walls crossed on the geodesic path, both for the ship and office environments. From that aspect, we can see that for the average link with average parameter values, the impact of walls crossed on the direct path in the office environment have a significantly larger impact on PL than the walls crossed on the geodesic path.

### 4.2. Errors

The traditional floor-map-based models have difficulty predicting PL in the ship environment as the distance between TX and RX antennas grows larger. This is again due to improper valuation of the wall crossings. Surprising however is the comparable performance of the ITU-R P.1238 model to the other traditional multi-wall models in this environment. It does not consider walls, but rather only whether or not links are in LOS or NLOS and uses model coefficients prescribed by the model itself. For the ship environment, it is clear that a pathfinding model is a must. The GPM prediction errors show an MAE of 4.79 dB, with a high coefficient of determination (R${}^{2}$) of 0.83, whereas the traditional floor-map-based models perform comparably among each other with an average MAE of 8.95 dB and an R${}^{2}$ value of 0.41.

The large under- and over-estimations in the ship environment from models such as AWM and COST231 stem directly from not being able to find a path. Especially in an environment with such a low PL exponent and at the same time, metal walls that attenuate significantly if TX and RX are not in the same path-connected space and a wall that divides path-disconnected spaces must be crossed. The significance of path-connectedness becomes apparent in the differences in wall loss coefficients from the GPM. This is however not something the AWM, COST231, or ITU-R P.1238 (site-general version) models can account for. For these traditional floor-map-based models, it’s near impossible to deal with the more prominent interaction losses as opposed to the very small distance losses. We would like to note that the availability of wall types would not solve this problem for the traditional floor-map-based models, because the wall types in the ship environment are all similar to each other. The important consideration of path-connectedness upon wall crossings is a necessary one, given an environment characterized by a low PL exponent.

However, in the office environment, we see that the average wall models perform significantly better than in the ship environment, although not better than the GPM. The GPM predictions show an MAE of 6.16 dB and an R${}^{2}$ value of 0.77. AWM and COST231 predictions have an MAE of 7.93 dB and an R${}^{2}$ value of 0.74. The ITU-R P.1238 (site-general version) model has the largest prediction errors and the model would need a PL prediction offset value far from zero. We would like to note however that the ITU-R P.1238 (site-general version) model does not consider walls, but rather applies different terms and coefficients based on environment type and whether or not the TX and RX antenna are in LOS.

## 5. Conclusions

In this research, we presented a novel path loss model and tool for predicting path loss. The model only requires the input of a simple binary 2-dimensional floor map, without the need for additional wall type information. The model is based on the use of shortest constrained distance, or geodesics, to find paths within path-connected spaces. Crossing walls to path-disconnected spaces is done systematically based on a heuristic. We presented an algorithm that automatically determines the parameters used in the model. Furthermore, the model was validated using path loss data from a considerably large measurement campaign in two vastly different environments, one being the main cabin floor in the superstructure of a ship and the other in an ordinary office environment. The path loss prediction results of the proposed Geodesic Path Model are mean absolute errors as low as 4.79 dB and a standard deviation of the error of 3.63 dB in the ship environment and a mean absolute error of 6.16 dB and standard deviation of 4.55 dB in the office environment. The most significant improvements are observed in the ship environment, an environment characterized by its low path loss exponent (1.15, as opposed to 2 for free space). We also find the importance of making a distinction between the crossing of walls that divide path-disconnected spaces as opposed to the crossing of walls that reside in the same path-connected space.

### Future Work

Validating our Geodesic Path Model with in-field measurements, not only allows us to have a better input for our active device-based localization efforts, but it opens the gate to other sensing objectives such as our device-free crowd or people sensing efforts. If determined propagation paths are reliable to a certain degree, then time-variable PL changes along those paths can facilitate the detection of people counts and presence in complex environments, whereas radio-frequency based device-free people counting traditionally requires a mostly unobstructed line-of-sight between TX and RX antennas.

We are interested in the reliability of the model in higher frequencies such as the millimeter-wave bands. At higher frequencies, the reflected signals become more diffuse and diffraction losses increase as well. In order to compensate for the incurred losses, directional antennas are used. This specific working of antenna directionality will need to be implemented in the path loss parameter determination tool.

## Figures and Tables

**Figure 1 sensors-22-04903-f001:**
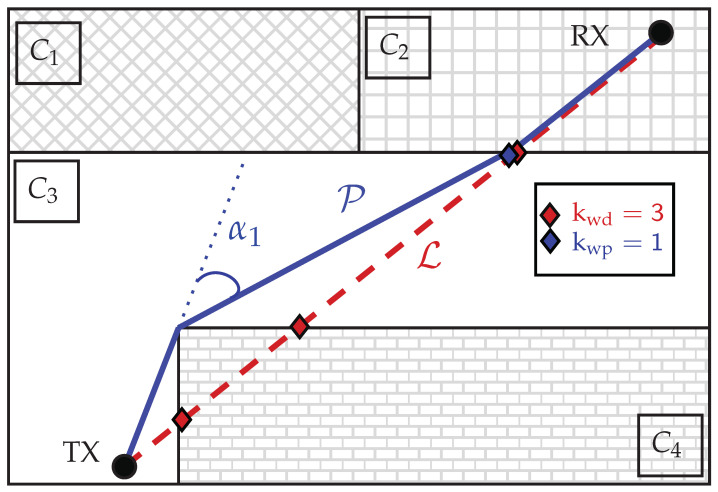
The blue line is the concatenated geodesic path P from TX to RX. Along it, the blue and red diamonds represent the points at which walls are crossed along the geodesic path and direct path L respectively. The number of walls crossings are enumerated by $k_{\mathrm{wd}}\;\mathrm{and}\;k_{\mathrm{wp}}$. The direction changes along the path P are indicated by $\alpha_{i},i\in \left\{0,\ldots ,\;\mathrm{corner}\;\mathrm{count}\right\}$. The distinct path-connected spaces are indicated by $C_{j},j\in \left\{1,\ldots ,J\right\}$.

**Figure 2 sensors-22-04903-f002:**
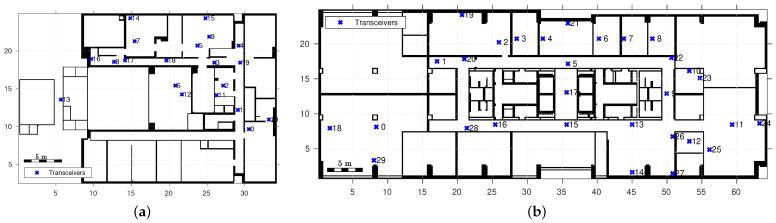
Floor map of the measurement environments and wireless sensor network deployment. The ship environment is show in (**a**) and the office environment in (**b**).

**Figure 3 sensors-22-04903-f003:**
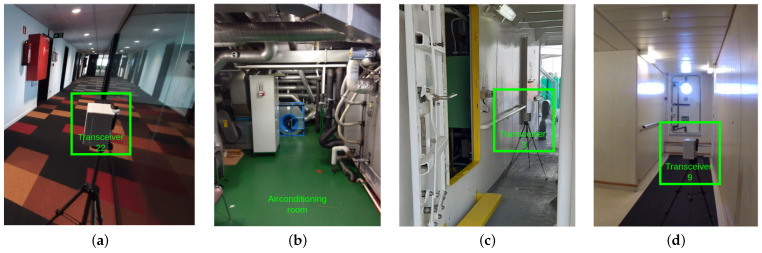
This figure shows the measurement environments. The office environment is a typical looking office, with a mix of unknown wall types of all sorts in (**a**), of which the location corresponds to that of transceiver 22 in Figure 2b. The ship environment is highly metallic one. The air conditioning room in (**b**) corresponds to the location of the transceivers 6 and 12 in Figure 2a. The location of transceivers 3 and 9 in (**c**,**d**) can also be referred to in Figure 2a.

**Figure 4 sensors-22-04903-f004:**
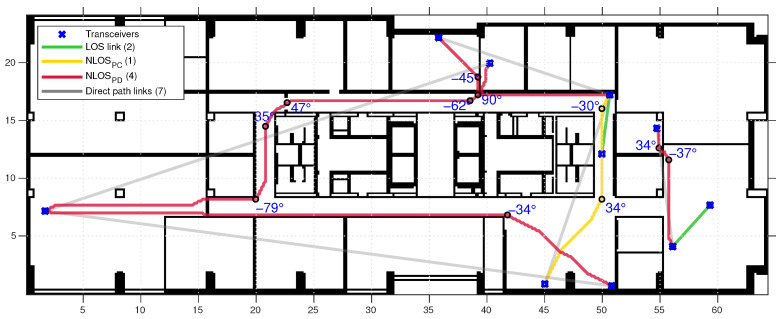
A visualization of the parameters produced by the algorithm, showing direction changes along the geodesic paths expressed in degrees, wall crossings by the direct paths and wall crossings by the geodesic paths. A randomly selected subset (approximately 3% of all links) is generated for visualization.

**Figure 5 sensors-22-04903-f005:**
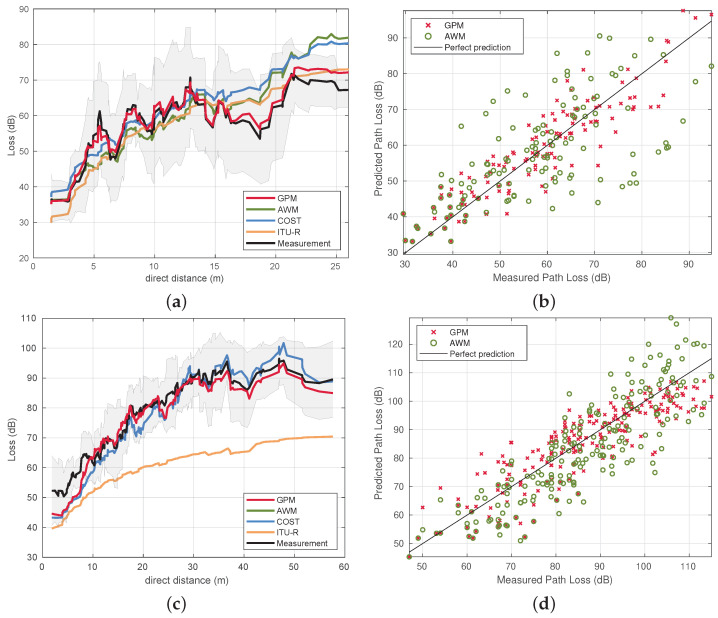
PL is outlined with respect to the direct path distance between TX and RX for all the considered models in (**a**) for the ship environment and in (**c**) for the office environment. A comparison between measured and predicted PL is made visible more explicitly in (**b**) for the ship environment and in (**d**) for the office environment.

**Figure 6 sensors-22-04903-f006:**
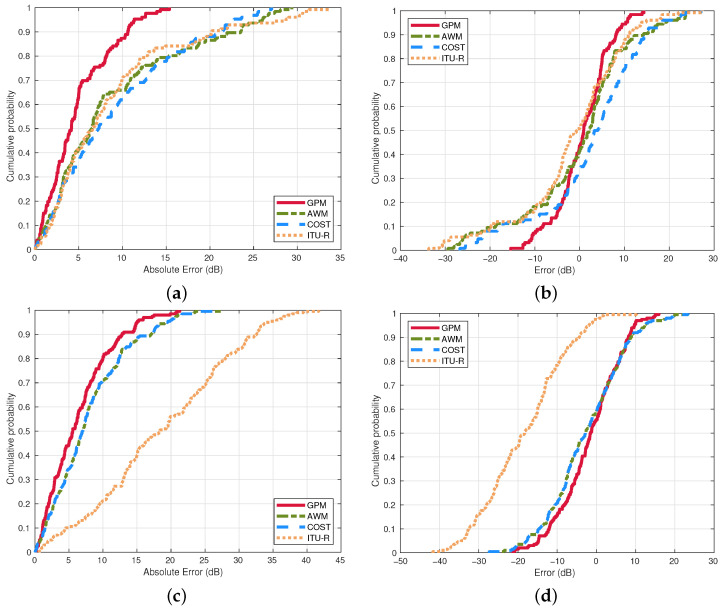
CDF plots of the PL estimation errors. The CDF of the absolute errors is shown in (**a**) for the ship environment and in (**c**) for the office environment.The CDF of the signed errors is shown in (**b**) for the ship environment and in (**d**) for the office environment.

**Figure 7 sensors-22-04903-f007:**
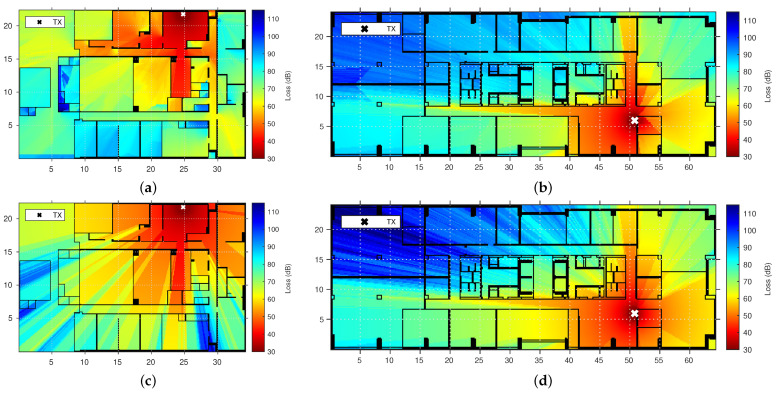
Predicted GPM PL is presented in (**a**) for the ship environment and in (**b**) for the office environment. For the purpose of comparison, the same is provided for the next best performing traditional model, AWM, in (**c**) for the ship environment and (**d**) for the office environment. We can see larger differences between GPM and AWM in the ship environment than in the office environment. The AWM considers walls and distance as the only path loss causes, while the GPM takes diffraction and differences in path-connectedness of wall crossings into account as well.

**Table 1 sensors-22-04903-t001:** Collected number of PL samples per link per measurements environment. A split is made between distinct links used for training and evaluation.

	Training Measurements		Validation Measurements	
	Link Count	Samples per Link	Link Count	Samples per Link
Ship	45	463	90	248
Office	63	121	210	89

**Table 2 sensors-22-04903-t002:** GPM, AWM and COST231 model coefficients for 868 MHz in a ship and office room environment, fitted using a multivariate non-linear least squares algorithm. The bold values indicate the coefficient estimates, accompanied by the lower limits (LL) and upper limits (UL) for the 95% confidence intervals (CI).

		GPM			AWM	COST231
		$L_{\mathrm{wd}}$	$L_{\mathrm{wp}}$	$L_{\alpha }$	$L_{\mathrm{wd}}$	$L_{\mathrm{wd}}$
Ship	Estimate	0.5588	17.79	9.6895	4.8137	2.9484
	95% CI [LL]	0.5357	17.5993	9.5737	3.9177	2.071
	95% CI [UL]	0.5819	17.9806	9.8053	5.7096	3.8258
Office	Estimate	2.2929	3.6716	4.5151	3.09	3.09
	95% CI [LL]	2.2628	3.2456	4.3487	2.7307	2.7307
	95% CI [UL]	2.323	4.0977	4.6814	3.4472	3.4472

**Table 3 sensors-22-04903-t003:** Statistical error analysis summary for the GPM, AWM, COST 231 and ITU-R P.1238.

	Model	MAE (dB)	$\sigma_{\left|\epsilon \right|}$ (dB)	$|\epsilon_{\max}|$ (dB)	RMSE (dB)	R^2^
Ship	GPM	4.79	3.63	16.38	6.0	0.83
	AWM	8.88	7.80	29.37	11.79	0.43
	COST 231	9.19	7.17	27.07	11.64	0.41
	ITU-R P.1238	8.79	7.83	33.65	11.75	0.38
Office	GPM	6.1637	4.55	20.783	7.66	0.77
	AWM	7.93	5.61	28.19	9.69	0.74
	COST 231	7.93	5.60	27.33	9.7	0.74
	ITU-R P.1238	18.73	9.92	41.72	21.19	0.67

## Data Availability

Not applicable.

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
