# Peer review of "Geodesic Path Model for Indoor Propagation Loss Prediction of Narrowband Channels"

_sensors, 2022, doi:10.3390/s22134903_

Round 1
Reviewer 1 Report
A new path model GPM that is presented by the authors is so interesting and the paper presentation is so well. But I think it needs to be revised before the acceptance with the following points.
1. The authors argue that their PL model in (5) is so simple “computationally light weight PL model” compared to the other PL models including AWM and COST231. But I guess the presented model looks more complex than the other ones since it needs to come up with four coefficients Lwd, Lwp, Lwp, La, while just one or two coefficients are enough for AWM or COST231 (see Table 1 and (2),(3)). What I mean is that derivation of those coefficients is so heavy. What do you think about my opinion?
2. Generally, one PL model that applies to one environment is also applicable to the other (ship or office) environments. But the presented PL model in (5) looks very specific so I am doubting that it is not well applicable for different environments. What is your opinion?
3. PL models including AWM, COST231 typically have a frequency parameter that is arrangeable but (5) has not. Why? Specifically, the presented PL model in (5) is just targeted for the DASH-7 with a frequency = 868M but DASH-7 has also covered the other two frequencies: 433M and 915M. Then, (5) is not applicable to 433M and 915M, is it?
Author Response
Dear Reviewer, please see the attachment.
I have attached the revised manuscript to the responses for your convenience.

Reviewer 2 Report
Paper proposes a propagation loss model for indoor, based on geodesics. It is complete, well structured and very well written. It has almost no typos.
The paper does not have a state-of-the-art section, and analysis of existing related work is almost nil. As a result, the number of references is very low. Research consists of adding knowledge to existing knowledge, but for that, we must carry out a survey of what has already been done, and under what conditions.
The technical component of the investigation consisted of the mathematical formulation and consequent simulation of a propagation loss prediction method, whose permissions are not completely defined. In fact, the formulation refers to the internal environment of a ship and the consequent difficulties associated with the fact that the walls are metallic, but it is never clearly defined which metallic walls are, much less which doors are closed and whether or not they are metallic.
When specifying the study environment, the ship's environment is then compared with an office environment, which is also ambiguous and without the possibility that the reader can repeat the experiments and verify the results. The plan does not define if there are elevators, where the pillars are and what the office walls are like.
Finally, and also with little value, the results of the propagation loss model are compared with other models, without measuring anything in the concrete scenario.
Your model may have fantastic efficiency, but you have to check it under real conditions. You have to guarantee repeatability to your experiences and without ambiguity (which of the walls are metallic or not, the doors are open or closed).
The references, besides being few, are mostly outdated. Only 35% of references are less than 5 years old.
The typo I found:
- line 273 “”than the GPM. the"
Author Response

(The authors gave the same response as above.)

Round 2
Reviewer 2 Report
Paper was considerably improved in my opinion, specially in terms of defining how experiences were conducted, and allowing the reader to try to repeat the experiences and to confirm the results. That component was very successfully addressed and it is OK.
The issue of missing related work component has not been better resolved. It is true that the author has described a set of recent, relevant papers describing related work, but he does not make any judgment as to how the works relate to his own work, what the flaws are, and in particular what the added value of his own work.
What did they did that does not apply to you scenario? What did you did better than them? What is your contribution to the state of the art?
I believe you can do it easily and that it would improve the paper, therefore for you final version, please write a set of comments in the text of the section, which will allow us to understand the limitations of each of the works for your case study.
Author Response
Dear Reviewer, please find attached our responses to your feedback. We have concatenated the manuscript for your convenience.
